# A Bio-Based Render for Insulating Agglomerated Cork Panels

Francesco Barreca [1],*, Natale Arcuri [2], Giuseppe Davide Cardinali [1] and Salvatore Di Fazio [1]

1    Department of Agriculture, Mediterranean University of Reggio Calabria, 89122 Reggio Calabria, Italy;
     gcardinali@unirc.it (G.D.C.); salvatore.difazio@unirc.it (S.D.F.)
2    Department of Mechanical, Energy and Management Engineering (DIMEG), University of Calabria,
     87036 Rende, Italy; natale.arcuri@unical.it
*    Correspondence: fbarreca@unirc.it; Tel.: +39-0965-1694215

**Abstract:** Natural and bio-based thermal insulation materials play an important role in the lifecycle impact of buildings due to their influence on the amount of energy used in indoor temperature control and the environmental impact of building debris. Among bio-based materials, cork is widespread in the Mediterranean region and is one of the bio-based materials that is most frequently used as thermal insulation for buildings. A particular problem is the protection of the cork-agglomerated panels from external stress and adverse weather conditions; in fact, cork granulates are soft and, consequently, cork panels could be damaged by being hit or by excessive sun radiation. In this study, an innovative external coat for cork-agglomerated panels made of a blending composite of beeswax and rosin (colophony) is proposed. The performance of this composite, using different amounts of elements, was analysed to discover which mix led to the best performance. The mix of 50% beeswax and 50% rosin exhibited the best performance out of all the mixes. This blend demonstrated the best elongation and the lowest fracture density, characteristics that determine the durability of the coating. A performance comparison was carried out between cork panel samples coated with lime render and beeswax–rosin coating. The coating of beeswax and resin highlighted a detachment value about 3.5 times higher than the lime plaster applied on the side of the cork.

**Keywords:** bio-based materials; building insulation; agricultural residual; beeswax; rosin; cork

## 1. Introduction

The most important European environmental challenge up to 2050 is to pursue the goal of becoming "climate neutral". The European commission (EC) in 2020 promulgated a specific action plan called the "Green Deal" to make the EU's economy sustainable. To reach this target, the EC requires action by all sectors of the European economy, including investment in environmentally-friendly technologies, support for industry to innovate and ensuring buildings are more energy efficient [1]. In fact, in developed countries, the building sector (buildings and related services) is responsible for about 40% of total energy use [2]. An increase in energy efficiency, especially in the building sector, has become a basic requirement [3] (Table 1).

**Table 1.** Specimen coating types.

| Blending Type | Beeswax (%) | Rosin (%) |
|---|---|---|
| I | 67 | 33 |
| II | 50 | 50 |
| III | 33 | 67 |
| IV | 23 | 77 |

At the same time, to overcome these challenges, Europe needs a new growth strategy, with agriculture and the environment as the top priorities among its economic and political aims [4]. For these reasons, natural and bio-based thermal insulation materials play

an important role in this challenge because of their influence on the energy required to maintain desired interior temperatures and their environmental impact on the embodied energy of the building [5–7]. The use of bio-based materials allows us to achieve some important aims [8–12]:

(1) to construct safer buildings and make interior environments healthier for humans to live in;
(2) to limit the amount of landfill created by demolished buildings, since bio-based materials are often biodegradable;
(3) to develop agricultural and ecological production, both of which create income for farmers and increase the amount of $CO_2$ absorbed, since most bio-based materials are derived from agricultural residuals.

Among bio-based materials, cork is widespread in the Mediterranean region; it is renewable and is widely and conveniently used for building thermal insulation. Cork features a high thermal resistance thanks to its elastic closed cell wall structure, which is a large part of its apparent volume [13–15]. The cell walls are composed of a waxy substance that is highly impermeable to gases and water. Another characteristic of cork insulation is its capacity to cope with significant thermal variations. The lifecycle energy of cork (embodied and operational energy) is less than half of that of other conventional insulators (expanded polystyrene, extruded polystyrene, rock wool and glass wool) [16]. Typical thermal conductivity values for cork are between 40 and 50 ($\times 10^{-2}$) $Wm^{-1} K^{-1}$ [17]. The production of agglomerated panels is the result of a process that recycles waste and residual cork into raw material. More than 75% of cork plank used to produce bottle stoppers in the world, after the production process, becomes residue [17–19]; furthermore, a large amount of waste cork is derived from forest cleaning and pruning, and from waste selection. This material is milled to obtain so-called cork granulate, which is used to produce agglomerated panels. These have been utilised in the building sector as insulating panels in various versions, depending on the binder used, gradation, and density. The type of binder used (urethane, melamine, or phenolic resins) to agglomerate the granulates influences their eventual mechanical and thermal behavior. A special building method has been developed in recent years by using the resin of cork (suberin) to bind the granules [20]. To soften the suberin, cork granules are overheated using high-frequency ultrasonic waves [21]. In comparison to the other insulation materials present in the market (EPS, XPS, PU, etc.), the insulation cork board features the lowest carbon footprint: 116.229 kg $CO_2$ equivalent per $m^3$ of cork board [14]. If we consider the carbon footprint for the whole cork sector by taking into account all the relevant production stages—from cork oak forest management, through the manufacturing processes to product distribution, use and end-of-life—it can be observed that the manufacturing stage, in our case including the transportation of raw cork and panel agglomeration, has the greatest impact on the environment because of the methane and carbon dioxide released from the combustion and decomposition of the biological material and from the combustion of fossil fuels. Nevertheless, the significant biogenic carbon emissions determined by the manufacturing process are largely compensated by the much greater carbon retention assured by age-old cork-oak forests. Moreover, [16] the recent political movement towards the promotion of the acquisition of local raw cork to reduce the transportation distance for the manufacturing and to make the production process more efficient will help to increase the competitiveness of the product. For all these reasons, cork panels are considered a sustainable and natural solution to the thermal insulation of buildings. Their utilisation is suitable not only as insulation against cold but also, thanks to their specific thermal capacity (1974.70–5467.50 kJ/kg) [22,23], as protection against hot temperatures. In fact, high specific thermal capacity material can delay and minimise indoor peak temperature by anti-phasing with outdoor temperature, and reduce the risk of summer overheating [16]. The application of cork agglomerated panels to the insulation of building walls involves different technical problems. The main problem is the protection of the panel from external stress and adverse weather conditions; in fact, the cork granulates are soft and the panel could be damaged by being hit or by excessive

sun radiation. Different solutions could be applied, such as coating the face with mortar lime [24] or green epoxy resins [18], thermo-shield coat layers [25], gypsum plaster [26], etc. In this study, an innovative external coat for cork agglomerated panels made of a blending composition of beeswax and rosin (colophony) is proposed. The performance of this composition, using different amounts of elements, was investigated to discover which mix led to the best performance. A performance comparison was carried out between cork panel samples coated with lime render and beeswax–rosin coating.

## 2. Materials and Methods

The coat materials investigated were as follows.

### 2.1. Lime Render

The outdoor side of cork panels is frequently protected against the effects of adverse weather by means of a lime coating layer (Figure 1). The most frequently utilised lime plaster is made from a binder of hydraulic lime mortars (EN 459-1) and river sand. The following materials were used to make the lime plaster: (1) a binder of hydraulic lime (NHL3.5), with a density of 1250 kg/m$^3$ and particle sizes of <200 µm; (2) river sand (silica and carbon), with a density of 2454 kg/m$^3$ and mean particle size of 0.489 mm; and (3) potable water. A formulation of mortar was developed by mixing one part of sand with three parts of NHL in weight and adding 0.3 liters of water to each kilo of mixture. A workability test of the hydraulic lime mortars was performed following the guidelines of the European standard EN 1015-3 [27].

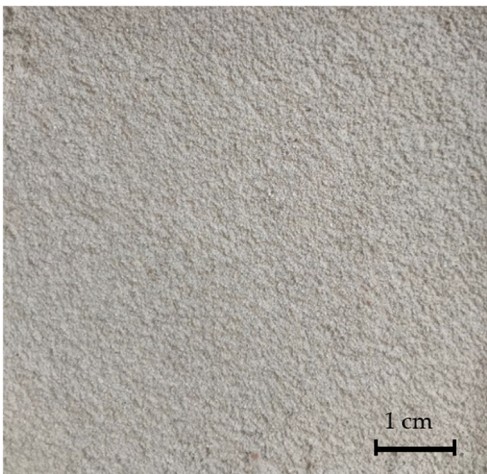

**Figure 1.** Lime plaster coating.

### 2.2. Beeswax-Rosin Coating

A natural resin named rosin (colophony) was used for the blend. Rosin (Figure 2b) is a resinous exudate of oleoresins from various species of pine trees with a melting point of 100−150 °C [28]. It is produced by heating the liquid resin to vaporise the volatile terpene components. The world's annual production of beeswax is estimated at about 1,270,000 tons [29]. The chemical structure is based on a hydrophobic backbone with hydrophilic attachments of carboxyl groups [28]. It features an average low molecular weight of 400 Da [30]. Rosin is a bio-sourced material known as a very good tackifier because of its very low surface tension when mixed with the correct solvent [31]. However, at room temperature, it remains very brittle, with a toughness in the order of magnitude of tens of kilopascals; this prohibits it from being included in most industrial applications in its pristine state. An amount of 90 wt % of Rosin is formed by a complex blend of diterpene-based acids with the empirical formula $C_{20}H_{30}O_2$. The other 10 wt % is a blend of esters, alcohols, aldehydes, and hydrocarbons. Beeswax is a bio-product used by bees to build the comb that forms the structure of their nest. The world's annual production of

beeswax is estimated at about 64,000 tons [32]. The beeswax used for the proposed coating is produced by the species Apis mellifera ligustica [33], which is the most popular honey bee in the world and is also widely bred in Italy, where beeswax is marketed for a wide range of uses (Figure 2) [34].

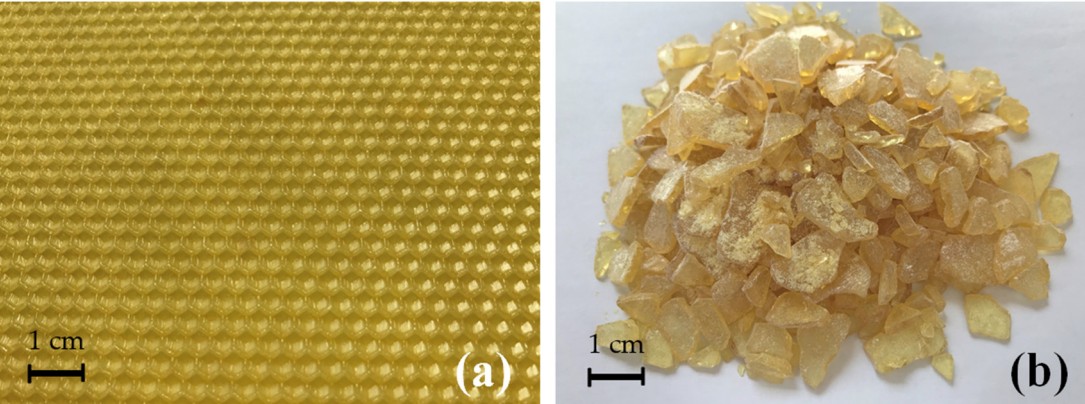

**Figure 2.** Materials for panel cork coating: (**a**) sheet of beeswax, (**b**) rosin.

Beeswaxes (Figure 2a) are known to feature very consistent compositions, similar to the following composition, given in wt %: hydrocarbons (14%), monoesters (35%), diesters (14%), triesters (3%), hydroxy monoesters (4%), hydroxy polyesters (8%), free acids (12%), acids esters (1%), acids polyesters (2%), free alcohols (1%), and various other compounds (6%) [33]. Due to its rich hydrophobic protective properties, beeswax is often used to coat wood furniture [35]. Beeswax is the substance that forms the structure of a honeycomb; the bees secrete wax to build the honeycombs, in which they store honey. Blends of beeswax and rosin (BR) form a partially crystalline material. Four distinct types of BR were mixed to make an agglomerated cork panel coating, and were subsequently evaluated (Table 1).

A mixture with a beeswax percentage of over 66% was not tested as it was extremely soft at room temperature. To prepare the blends, the beeswax and rosin were heated and melted together at a temperature of 100 °C for approximately 15 min, then mechanically mixed for about 30 min before they were used to prepare the samples for the test (Figure 3) [36]. Specific tests were designed to evaluate the most suitable beeswax and rosin blend, with reference to the requested performance of a cork panel coat. The performance was evaluated according to the following criteria: tensile strength, tensile elongation strain, surface hardness in use, hydrophobicity, and thermal stress. Compromise Programming (CP) [37] was the method adopted to compare the different coating blends and to obtain the best solution. CP employs the concept of distance to analyse multiple-objective problems. This distance is not limited to the Euclidean distance between two points but is used as a proxy to measure degrees of human propensity. CP selects a solution from a feasible set, on the basis of the solution's closeness to a hypothetic ideal point [37]. The ideal point represents the joint location of the individual maximum values of all the objectives, and the best compromise solution is the nearest solution to the ideal solution with the highest value objectives.

The best BR blending coating type was successively compared with the most diffused cork panel render made of mortar lime [38]. Various tests were performed to characterise the coating in terms of: tensile strength, hardness, hydrophobicity, fracture density, and glossiness. Next, two important tests were conducted to evaluate the suitability of the proposed solution: the detachment strength from the cork panel substrate and the emissivity thermal radiation of the material's surface.

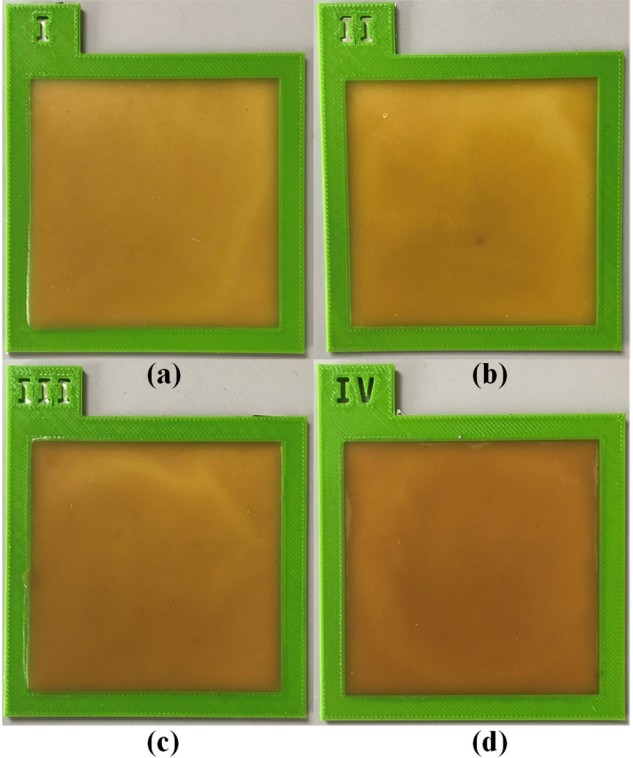

**Figure 3.** BR specimens: (**a**) Type I—67% beeswax, 33% rosin, (**b**) Type II—50% beeswax, 50% rosin, (**c**) Type III—33% beeswax, 67% rosin, (**d**) Type IV—23% beeswax, 77% rosin.

### 2.2.1. Tensile Strength Test

Tensile strength and elongation are two important characteristics of cork panel coating [39]; in fact, the coat has to allow for, and follow, the deformation and elongation of the agglomerated cork panel when it is exposed to natural weather conditions without detachment and damage. For each type of blend, five specimens were prepared according to ASTM D638 standards [40]. The specimens were prepared by means of a specific frame to contain and form the melting blend material (Figure 4).

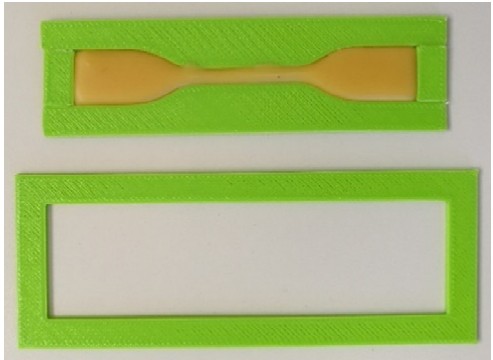

**Figure 4.** ASTM D638 type IV specimen.

The tensile stress tests were carried out by means of a dynamometer with external load cell digital force gauges, the SamaTools SADFGE-P (Figure 5).

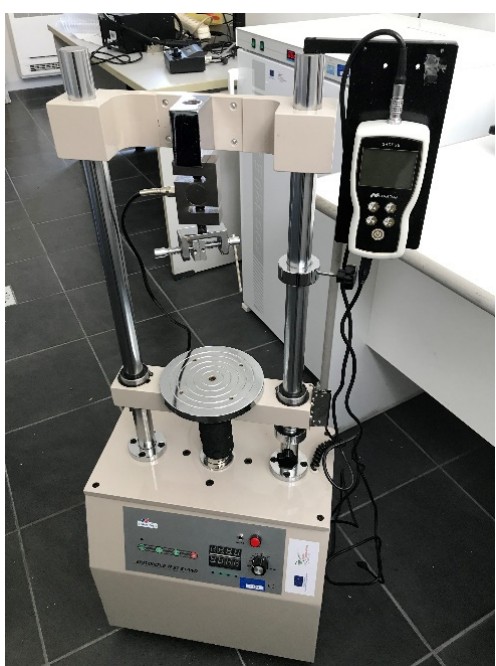

**Figure 5.** Double-column motorised Samatool-SAHDV-10K.

A tensile speed of about 5 mm/min was applied to the apparatus. The tensile stress and the elongation data were recorded for each specimen and the average tensile stress curve of the four types of blend of the specimens was obtained (Figure 6) (Table 2).

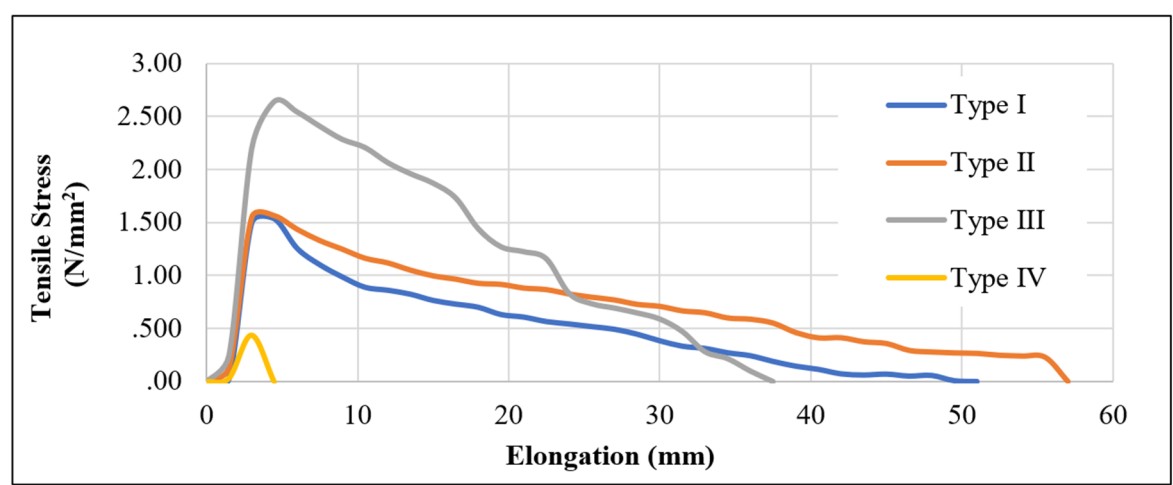

**Figure 6.** Tensile stress/elongation curves: Type I—67% beeswax, 33% rosin; Type II—50% beeswax, 50% rosin; Type III—33% beeswax, 67% rosin; Type IV—23% beeswax, 77% rosin.

**Table 2.** Yield tensile strength average values of the specimens.

| Yield Tensile Strength (N/mm$^2$) | Type I | Type II | Type III | Type IV |
|---|---|---|---|---|
| | $1.54 \pm 0.22$ | $1.55 \pm 0.33$ | $2.64 \pm 0.49$ | $0.44 \pm 0.09$ |

The highest elongation value (69 mm), in the plastic phase, was recorded for the Type II specimen. The lowest elongation was recorded for the Type IV specimen. A break without elastic or plastic phases was observed; this is typical of its fragile behavior.

The Type IV specimen recorded an average value on the peak stress measurement lower than that measured for the other types. In fact, less than 10% of beeswax in the mixture determined the fragile behavior of the specimen.

The Type III specimen recorded the highest tensile strength value but not a high elongation value. A high elongation capacity value of the coating, such as that of the Type II specimen, allows high deformation of the panel without losing its protective function.

### 2.2.2. Hardness

Hardness is a resistance indicator of the coating against scratches, wear or deterioration due to environmental conditions.

The measurements of the hardness of the surface were conducted in accordance with ASTM D2240 type D scale [40]. For each type of blend, a square specimen of 100 mm × 100 mm and 2 mm thickness, was prepared and stored in a laboratory climate for over one hour before testing. Five hardness tests were conducted for each specimen by means of a Shore D durometer.

The average hardness values are reported in Table 3.

**Table 3.** Hardness (Shore D) average values.

| Hardness (Shore D) | Type I | Type II | Type III | Type IV |
|---|---|---|---|---|
| | 30.91 ± 1.46 | 25.81 ± 1.83 | 33.37 ± 2.49 | 42.63 ± 2.29 |

Different polymorphic transformations occurred during the heating/cooling of the blend as a function of the rosin content. During use, cork panels are exposed to environmental conditions and to natural thermal cycles. For these reasons, the behavior of the coating to different cycles of heating and cooling was investigated. The specimens were stored for 12 h at 40 °C with 10% air relative humidity and for 12 h at −17 °C with 70% air relative humidity; each cycle was repeated four times. For each stage of the cycle, measurements of hardness were conducted to evaluate the behavior of the blends at different temperatures. The average hardness values after the last heating and cooling cycle are reported in Table 4 and Figure 7.

**Table 4.** Hardness (Shore D) average values—Hot/cold cycle.

| CYCLE | Type I | Type II | Type III | Type IV |
|---|---|---|---|---|
| Hot (Shore D) | 7.41 ± 0.76 | 25.21 ± 2.93 | 23.16 ± 1.96 | 26.03 ± 2.04 |
| Cold (Shore D) | 54.5 ± 2.33 | 42.77 ± 4.83 | 53.52 ± 4.41 | 62.64 ± 1.85 |

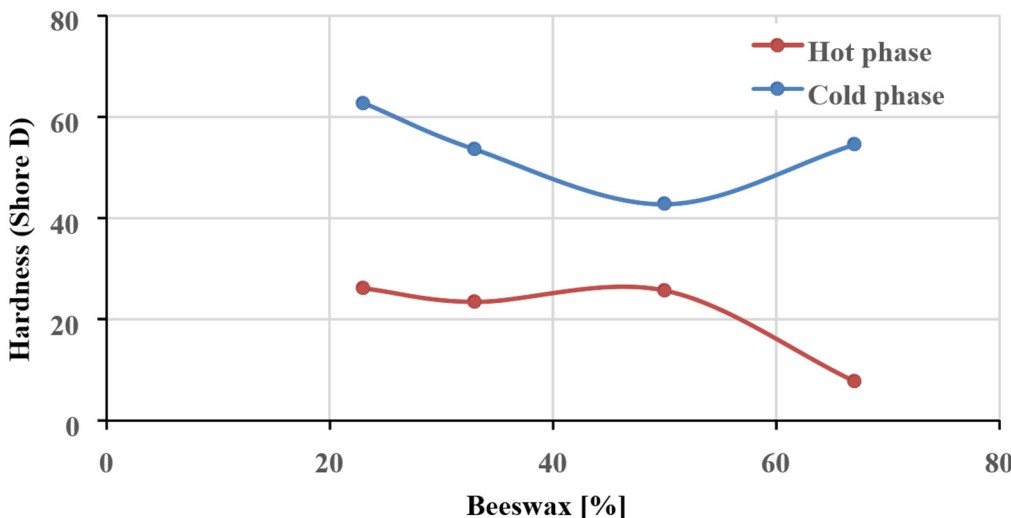

**Figure 7.** Hardness (Shore D)—hot/cold phase.

It is important to note that previous studies highlighted that, for the elastomeric materials, there is an almost linear relation between the logarithm of elastic modulus and

the hardness scale. The $E_0$ for the different blend samples was calculated by means of Equation (1):

$$logE_0 = 0.0235S - 0.6403 \tag{1}$$

where S = Shore D + 50 80A < S < 85D.

### 2.2.3. Hydrophobicity

In general, two important surface parameters influence hydrophobicity: surface energy and texture [35]. Low surface energy and surface texture with micro/nanopillars are favorable to the enhancement of surface hydrophobicity. The surface hydrophobicity of the samples made with different BR was evaluated using the optical tensiometer Theta Lite product by Attension® (Figure 8).

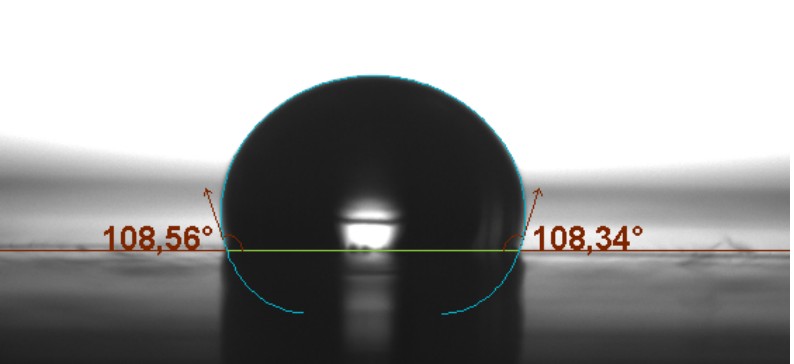

**Figure 8.** Measure of hydrophobicity angles by Theta Lite product.

The contact angles between the droplets of water and the sample surfaces were measured in accordance with ASTM D7490-13 [41]. The tests were conducted at a temperature of about 23 °C and at a 60% relative humidity (Table 5).

**Table 5.** Hydrophobicity value.

| Hydrophobicity (°) | Type I | Type II | Type III | Type IV |
|---|---|---|---|---|
| | 88.42 ± 5.94 | 107.31 ± 4.64 | 98.69 ± 4.81 | 96.44 ± 8.37 |

### 2.2.4. Fracture Density

An important characteristic of the coating is its fracture toughness and thermal shock resistance during use. Specific tests were conducted to evaluate the anti-fracture strength of the different types of blend investigated. In particular, for each specimen, with a surface of 10,000 mm$^2$, the total lengths of fracture lines (Figure 9) due to the durometer's indenter were measured.

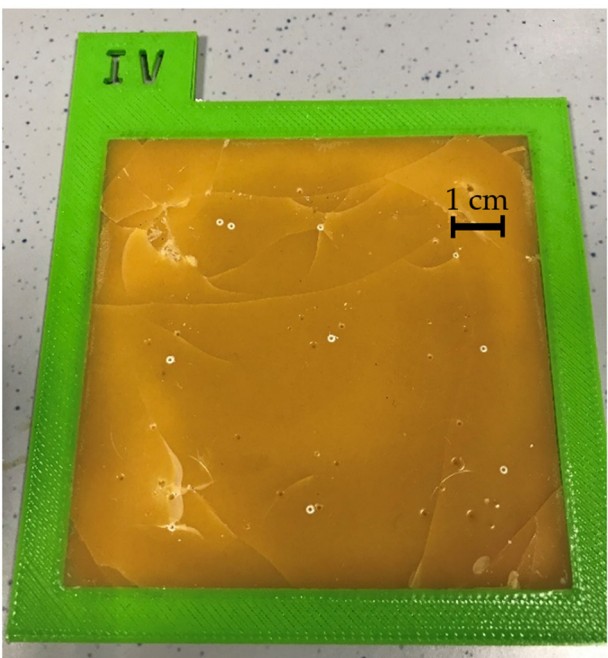

**Figure 9.** Fracture lines in specimen type IV.

The measures were conducted at the end of the thermal cycles to evaluate the thermal shocks that could produce an increase in brittleness [42]. The total fracture line lengths were correlated to the surface area of the specimen to calculate the values of fracture density (Table 6).

**Table 6.** Average values of fracture density.

| Fracture Density (mm·mm$^{-2}$·10$^{-4}$) | Type I | Type II | Type III | Type IV |
|---|---|---|---|---|
| | 5.21 ± 0.12 | 2.10 ± 0.13 | 43.29 ± 0.24 | 59.89 ± 0.24 |

### 2.2.5. Glossiness

Glossiness is the property of the appearance of smooth surfaces, which is associated with the surfaces' ability to reflect light in some directions more than others, giving rise to so-called highlights. The directions of specular reflection, where light is reflected at the same angle on the normal surface as the incident light, typically feature the highest level of reflectance (Figure 10).

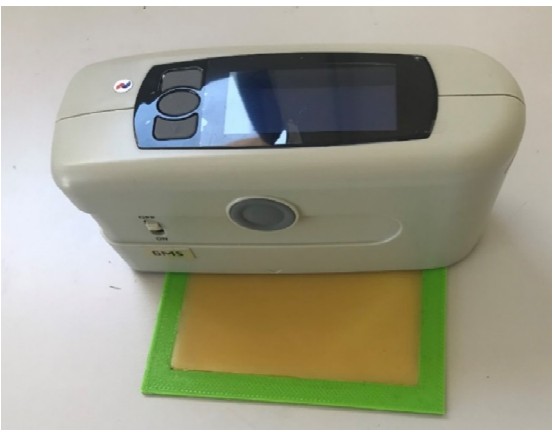

**Figure 10.** Glossiness specimen measure.

Materials with very high specular reflection, such as highly polished metals, tend to feature low emissivity of ambient infrared energy and to be less effective at emitting their own electromagnetic waves. For these reasons, a coating material with high reflectance improves the performance of cork panels with respect to the sun's radiation [43]. These measurements of the glossiness were conducted in accordance with ASTM D523 (Table 7).

**Table 7.** Glossiness values resulting from the test conducted in accordance with ASTM D523.

| Glossiness (GU) | Type I | Type II | Type III | Type IV |
|---|---|---|---|---|
| | $7.18 \pm 0.45$ | $6.01 \pm 0.45$ | $14.52 \pm 0.64$ | $44.72 \pm 4.25$ |

## 3. Results

The specimens with different BR performed differently on the tests. For this reason, it was necessary to conduct a specific procedure to highlight the best suitable solution. A specific comparison was carried out by means of the compromise programming method. The performance values were normalised using Formula (2). If a specimen had shown the value 1 for all the performances it would have been the best choice; therefore, this was defined as the "ideal" solution. Consequently, the BR that highlighted the minimum difference ('distance') to the "ideal" solution was the best alternative solution and was chosen (Table 7).

$$\bar{y}_i^j = \frac{y_i^j}{y_i^{max}} \tag{2}$$

where:

$\bar{y}_i^j$ is the normalised value of the *j*th mix type of the *i*th performance value

$y_i^j$ is the value of the jth mix type of the *i*th performance value

$y_i^{max}$ is the max performance of the *i*th value

$$Y^j = \left( \sum_{i=1}^{4} \left( \bar{y}_i^j - y_i^{max} \right)^2 \right)^{\frac{1}{2}} \tag{3}$$

where:

$Y^j$ is the distance of the *j*th mix type from the ideal solution

$\bar{y}_i^j$ is the normalised value of the *j*th mix type of the *i*th performance value

$y_i^{max}$ is the max performance of the *i*th value

The final values obtained by the use of the compromise programming method [3] are reported in Table 8.

**Table 8.** Programming method final normalised values.

| Specimen | TYPE I | TYPE II | TYPE III | TYPE IV |
|---|---|---|---|---|
| Ultimate tensile strength | 0.58 | 0.59 | 1.00 | 0.17 |
| Elongation max | 0.70 | 1.00 | 0.39 | 0.00 |
| Hardness | 0.87 | 0.71 | 0.87 | 1.00 |
| Hardness frost-defrost—COLD ($-17$ C) | 0.87 | 0.68 | 0.85 | 1.00 |
| Hardness frost-defrost—HOT ($+40$ C) | 0.29 | 0.98 | 0.90 | 1.00 |
| Hydrophobicity | 0.89 | 1.00 | 0.96 | 0.87 |
| Glossiness | 0.16 | 0.13 | 0.32 | 1.00 |
| Fracture density | 2.46 | 1.00 | 20.52 | 28.40 |
| E | 0.73 | 0.49 | 0.73 | 1.00 |
| Ideal point's distance | 1.93 | 1.17 | 19.54 | 27.43 |

A mix of 50% beeswax and 50% rosin exhibited the best performance of all the mixes, which was mainly due to the maximum elongation and the fracture density, which determine the durability of the coating. In fact, these two characteristics allow the coating,

when exposed to weather conditions, to deform without break. It is common to coat the agglomerated cork panels with a layer of mortar lime of about 3 mm, to protect the external surfaces when they are applied to the faces of a building's envelope. Recently, some criticisms about the environmental and health and safety performances of this solution have been highlighted, in addition to the reported defects during use [25,44]. For this reason, in this study, some properties in the use of the BR coating proposed were analysed. In particular, the coating's adhesion to the cork panel and the sun radiation reflectance property were investigated. It is very important to limit the emissivity of the surface and to improve the values of the solar reflectance in order to improve, in turn, the thermal performance of the layer of insulation. In recent years, interest in the manufacturing of advanced insulation panels using eco-friendly materials has grown. When these panels are employed as an external insulator, a shield coating is used in order to enhance the surface radiance features. It is also important to minimise the risk of a subsurface detachment during the thermal conduction via heat transfer. To achieve this, an adhesion test was conducted to evaluate the strength of the coating detachment of the face of the cork agglomerated panel. Because of the manufacturing procedure of the cork panel, the two faces of the cork panel could show different surface roughness depending on the diameter of the granules [45].

### 3.1. Coating Detachment Strength

Different national and international codes define the standard test methods for the determination in situ of the adhesive strength of rendering and plastering mortars to their substrate (ASTM C1860-20; BS EN 16602-70-13 and EN 1015-12, 1348, 1542, 12616-2, 13963, 14496, surface protection mortars EN 1504-2 EN 1542) [25,38,46,47]. The apparatus used for the test is a machine for the Pull-Off test. Essentially, it is a dynamometer fitted with a loaded cell. The direct tensile force is applied to the render by a circular pull-head plate made of stainless steel (with a diameter of $50 \pm 0.1$ mm and a thickness of 10 mm) attached by an adhesive-based resin. After the cure time of the render mortar, the samples must be cut out using a core drill device (up to a depth of about 2 mm within the support). These tests are not suitable for measuring the bond strength of the BR coating of agglomerated cork panels. The free fatty acids and the oleate esters contained in the beeswax make the bonding of any adhesive on the surface difficult. For the version of the test used in this study, a specific method was proposed. It consisted in applying the tensile force to the coating by means of a circular disk (with a diameter of $20 \pm 0.1$ mm and a thickness of 2 mm) immersed in the coating before it was spread on the surface. The bond strength of the coating sample of beeswax was applied by means of a double column motorised force gauges test stand, SAHDV—10K, and measured by high-precision digital force gauges, SADFG-P. The tensile velocity was 1.5 mm·s$^{-1}$. The sample was cut out using a core drill device (up to a depth of about 2 mm within the support) (Figure 11).

The analysed cork panels exhibited different textures between the two sides. One side exhibited a texture composed of larger cork granules (0.2–0.5 mm) and the other side featured a texture composed of smaller ones (0.05–0.2 mm). In fact during, the compression process, the smaller cork granules moved down to the base, and the larger granules rose. For this reason, the above tests were conducted on four samples. Specimens of 10 cm × 10 cm cork panel were coated, two with a layer 3 mm thick composed of a mix of 50% beeswax and 50% rosin, and two with a layer 3 mm thick composed of lime plaster. Both the different roughness cork panel specimen faces were tested (Figure 12).

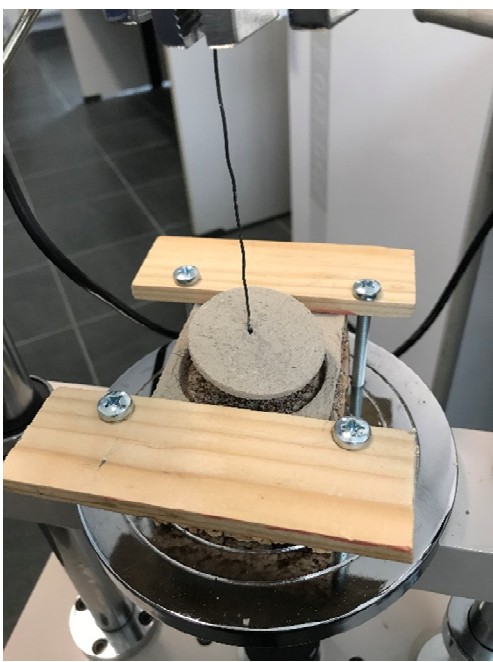

**Figure 11.** Coating detachment tests with double column motorised force gauges test stands SAHDV—10K.

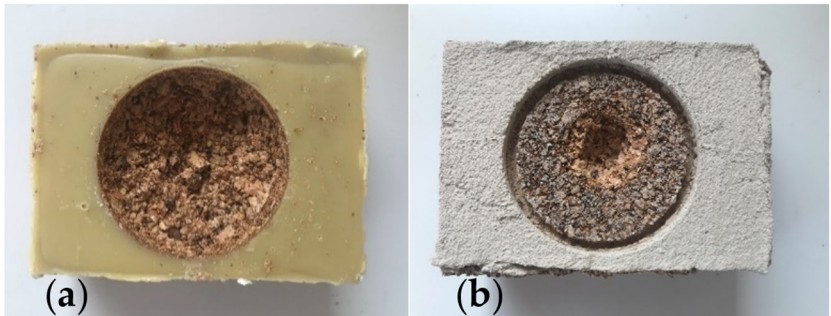

**Figure 12.** Coating detachment tests: (**a**) BR coating, (**b**) lime plaster coating, cork detached.

The coating was applied to each face of the cork panel specimen to evaluate the different bond strengths in relation to the surface roughness. The values are reported in Table 9 (Figure 13).

**Table 9.** Ultimate tensile strength.

| Coating | Tensile Stress Peak (N) |
|---|---|
| Lime plaster on fine cork granules | $145 \pm 1.78$ |
| Lime plaster on large cork granules | $130 \pm 1.23$ |
| Beeswax/rosin on large cork granules | $463 \pm 0.97$ |
| Beeswax/rosin on fine cork granules | $323 \pm 1.12$ |

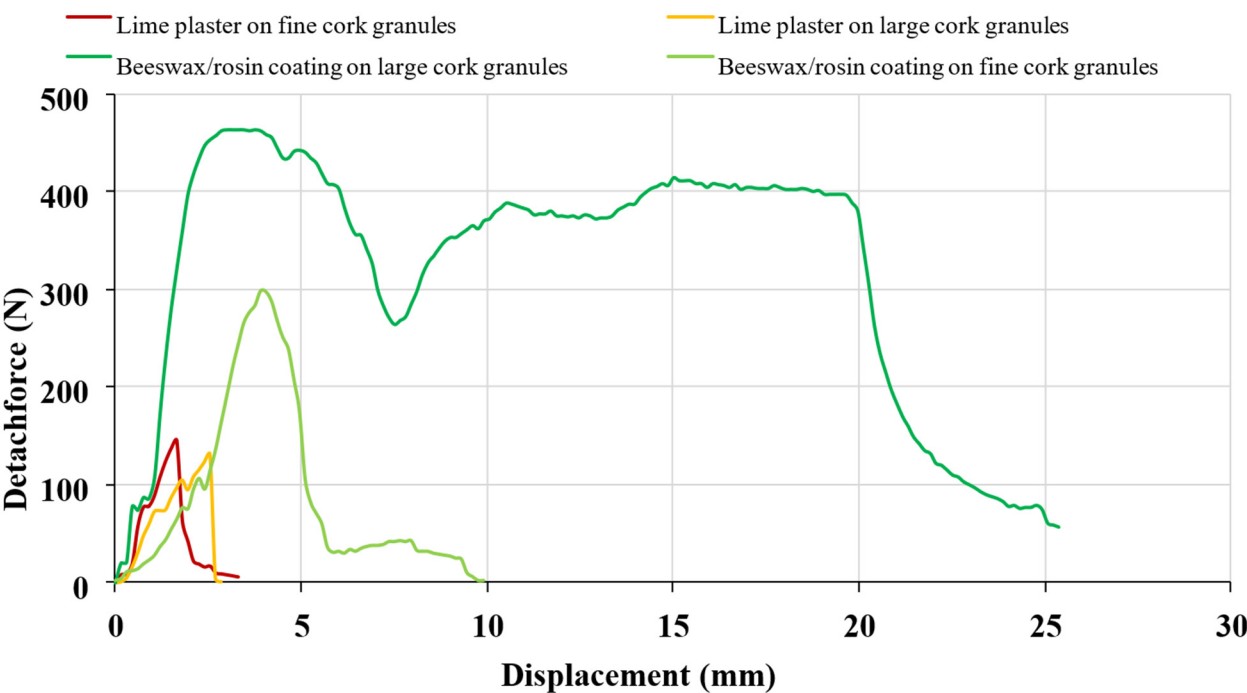

**Figure 13.** Detachment force-displacement curve of the coating on the cork panel.

It is important to note the approach to the two-coating detachment (Figure 14). The detachment of the BR coating was determined by the splitting of the cork granules of the substrate; for the lime plaster coating, however, the detachment was determined by the adhesion loss of the coating from the substrate.

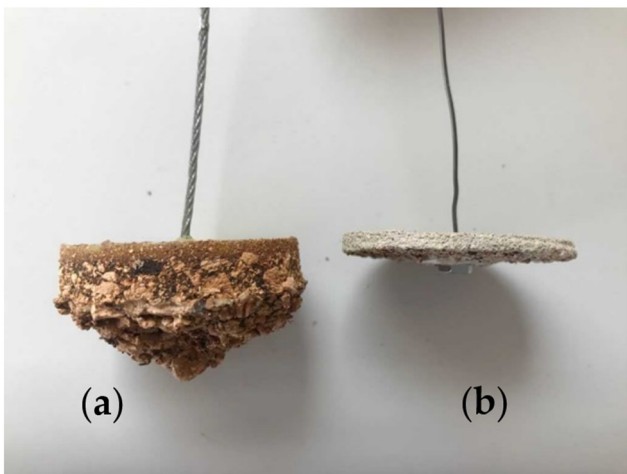

**Figure 14.** Coating detachment approach: (**a**) BR coating, (**b**) lime plaster coating.

*3.2. Emissivity Properties*

An important parameter of the external surface of the building envelope is the emissivity ($\varepsilon$) property, which defines the ability of a material to emit energy, and is strongly correlated with its surface characteristics [48]. Emissivity values can vary between 0 (perfect reflector) and 1 (perfect emitter) [49]. In hot climates, the building envelope should feature a low emissivity value to prevent the indoor environment from overheating. To measure the emissivity of materials, infrared thermography (IRT) was used [50]. For this purpose, four 30 cm × 30 cm, 3 cm thick cork panel samples were prepared. Two panels were of blond cork, one with fine-granule- cork and the other with large-granule cork on the side; one panel was of dark brown cork furthermore another panel was of blond cork

but with a 3 mm thick coating of BR type II. The emissivity value of the specimens was measured in accordance with ISO 18434. The reference emissivity material method was applied by means of an Infrared Camera (IRT) FLIR B 335 (Figure 15).

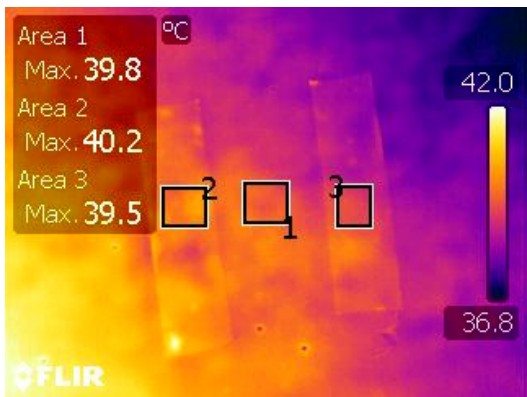

**Figure 15.** IRT emissivity of BR coatings.

The measurements were conducted in a dark room with a constant air temperature of 9.30 °C and a humidity of 61%. The samples were heated over 24 h by means of a fan jet oven. A black matte non-cloth tape with an $\varepsilon$ = 97 was applied to the cork panels' surface for reference. Using the camera software, the tape temperature was measured in order to consider its emissivity [51]. The temperature value was the average of the tape temperatures inside a predefined area. The surface emissivity was then adjusted by equating the average temperature of the tape and the average temperature of the surface near it, considering a similar area. The emissivity values were calculated with the average value of three measurements for each sample (Table 10).

**Table 10.** Coatings and cork emissivity values.

| Panel Surface | Emissivity Value ($\varepsilon$) |
|---|---|
| Side with fine cork granules | $0.62 \pm 0.03$ |
| Side with large cork granules | $0.93 \pm 0.02$ |
| Dark brown cork panel | $0.86 \pm 0.03$ |
| Coating of BR | $0.73 \pm 0.02$ |

## 4. Discussion

The results of the detachment strength test demonstrated that the specimens of cork with lime plaster and cork with BR behaved differently. In particular, the coating of beeswax and resin highlighted a detachment value around 3.5 times higher than the lime plaster applied on the side of the cork with large granules. The same detachment value was around 1.8 times higher than the lime plaster applied on the side of the cork panel with smaller granules. The way it detached was more evident; the wax resin coating stuck very firmly to the cork side, and the detachment was due to the disconnection of the cork granules, whereas the detachment of the mortar plaster was due to the failure of the coating to adhere to the cork base (Figure 14). The emissivity measurements conducted on the different cork panel types and surface textures produced further interesting results. The spectral emissivity over the thermal IR is a key property in determining energy transfer. The spectral emissivity for opaque coatings is given by (4)

$$\varepsilon\,(\lambda) = 1 - \rho(\lambda)$$

where:

$\varepsilon\,(\lambda)$ is the spectral emissivity at wavelength $\lambda$; and
$\rho(\lambda)$ is the reflectance at wavelength $\lambda$

In particular, BR coating improved the reflectance of the cork panels with large granules by more than 20% and that of the brown cork panels by more than 15%, but reduced the reflectance of the cork panels with small granules by about 29%. The low emissivity of the panel side with smaller granules was due to the presence of silica and other cork bark impurities in the granules mass used to form the panels. These reduce during the compression and vibration phase and accumulate the silica on the surface [52]. In this case, for the BR coating of BR, although the emissivity grows, a more homogeneous and opaque surface was created [53,54]. The colour of the surface was amber, and dependent on the beeswax clarity, which was due in part to how much the wax was filtered and in part to the type of flowers the bees had foraged. The final colour was due to the substrate colour and the thickness of the coating layer; the latter was conditioned by the roughness and planarity of the substrate. In this study, a thickness of 2 mm to cover all the agglomerate cork panel surface utilised for the tests was tested. The surface of the BR coating was smooth, and it was not possible to paint it with different colors. The addition of colour pigments to the tested blends, in order to obtain a specific color, will be explored in our future research.

## 5. Conclusions

The utilisation of green and bio-based materials in the building sector is increasing and, in the near future, it will be further encouraged by NextGenerationEU, the temporary instrument designed to boost the recovery and improve resilience following the COVID-19 crisis. It will be the largest stimulus package ever financed in Europe, aiming to make it a greener, more digital and more resilient continent. Green and bio-based materials are endowed with excellent properties and demonstrate good performance in use. It is important to investigate them to evaluate in a scientific manner their characteristics, performance values and potential uses, in terms of opportunities and limitations. At the same time, it is important to utilise bio-based materials in an appropriate way because incorrect use can limit their ecological properties. Cork panel coating with a BR layer, as highlighted in this study, is a good solution through which to refine the exposed side of the panel, to protect the panel against environmental deterioration, and to improve the ecologic footprint of building components.

The main findings of this study were as follows:

- A coating with a BR layer creates a more hydrophobic surface of cork panel insulation and also allows it to maintain the same insulation property in a wet environment.
- BR coating demonstrates good elongation and allows for the deformation of the substrate without it breaking.
- The BR layer features a high adhesion strength with the cork substrate because the composite penetrates inside the cork matrix.
- The BR layer demonstrates a high level of hardness and low brittle behavior in the context of high environmental temperature excursions.
- The BR coating diminishes the surface emissivity of the substrate, raising the reflectivity of the sun radiation.

Future studies will investigate important performance features of the coating in use, such as: thermal insulation properties, ageing, wearing, UV resistance, coupling durability between coating and base support, etc. For these aims, specific tests will be conducted on specific case studies on a real scale. The proposed coating could be applied to other cork components, such as: home furniture, boat components, tanks, food container, and for all these applications is necessary to conduct specific tests. Further studies should be carried out in the future to evaluate the performance of the coating when applied to other natural materials of vegetal origin (e.g., wood, bamboo, hemp) and mixed with other materials to improve its performance [55], given that its use in the building components of agricultural products, byproducts and waste can support the agricultural market and generate extra incomes for farmers particularly in disadvantaged regions [56,57].

**Author Contributions:** Conceptualization, F.B.; data curation, G.D.C.; investigation, F.B. and G.D.C.; methodology, F.B., S.D.F., and N.A.; software, G.D.C.; supervision, F.B., S.D.F., and N.A.; review and editing, S.D.F. and N.A.; writing—original draft, F.B. and G.D.C. All authors have read and agreed to the published version of the manuscript.

**Funding:** The research was funded by the project Proof of Concept 01_00052 included in the MIUR-PNR 2015–2020.

**Institutional Review Board Statement:** Not applicable.

**Informed Consent Statement:** Not applicable.

**Data Availability Statement:** Data sharing is not applicable to this article.

**Conflicts of Interest:** The authors declare no conflict of interest.

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
