# Peer review of "A Bio-Based Render for Insulating Agglomerated Cork Panels"

_coatings, doi:10.3390/coatings11121478_

Round 1
Reviewer 1 Report
The article is important for cognitive knowledge. One may wonder whether the authors should not make more comparisons of the various ratios of rosin to beeswax. However, I believe that the article brings essential knowledge to scientific discussion or practical application. It should be emphasized that cork waste is used for the production of this thermal insulation. Considering the increasingly popular principle of circular economy and the reproducibility of this raw material, this material is worth considering.
The article is written with great care. However, there are some minor bugs:
Table 6. We start the caption under a table with a capital letter. Please check line spacing throughout the article. Picture 13. We usually sign the drawing at the bottom. Figure 14, there is an unnecessary comma at the end of the caption. Quote (from page 2): It is important to ensure that the panels are sustainable and safe for the inhabitants of the building, and above all, that the cork panels are installed on the inside of the building envelope. If thermal insulation panels are installed on the inside of the wall, what is the vapor permeability of the partition? I wonder if it would not be correct to apply an ecological analysis of this material, for example LCA. In my opinion, this would increase the interest of the readers.
Author Response
Response to Reviewer Comments
Dear reviewer,
Thank you very much for giving us the sincere advices to improve our paper. We had carefully read your comments and revised the manuscript accordingly. Noticeably we have the following changes:
Reviewer #1
R: The article is important for cognitive knowledge. One may wonder whether the authors should not make more comparisons of the various ratios of rosin to beeswax. However, I believe that the article brings essential knowledge to scientific discussion or practical application. It should be emphasized that cork waste is used for the production of this thermal insulation. Considering the increasingly popular principle of circular economy and the reproducibility of this raw material, this material is worth considering.
The article is written with great care. However, there are some minor bugs:
A: We thank the reviewer for the appreciation of our work. The minor bugs detected, references in particular, have been amended according to reviewer’s suggestions
R: Table 6. We start the caption under a table with a capital letter. Please check line spacing throughout the article. Picture 13. We usually sign the drawing at the bottom. Figure 14, there is an unnecessary comma at the end of the caption.
A: These aspects have been revised according to the reviewer’s corrections and suggestions.
R: Quote (from page 2): It is important to ensure that the panels are sustainable and safe for the inhabitants of the building, and above all, that the cork panels are installed on the inside of the building envelope. If thermal insulation panels are installed on the inside of the wall, what is the vapor permeability of the partition?
A: We have deleted the sentence because could cause some confusion, since in this study we considered the cork panel fixed to the external the building envelope
R: I wonder if it would not be correct to apply an ecological analysis of this material, for example LCA. In my opinion, this would increase the interest of the readers.
A: We agree in that this is an interesting and necessary theme of investigation. Presentlly there are not enough specific LCA data sets or environmental impact studies about rosin (colophony) and beeswax. We plan to address our future research also in this direction.
Reviewer 2 Report
The authors of the article entitled "A Bio Based Render for Insulating Granulated Cork Panels" proposed and tested an innovative external coat for the cork agglomerated panels made of a blending composite of beeswax and rosin (colophony).
At the outset, I have doubts whether the word composite should be used in this case, or rather composition? Although it is difficult to find one consistent definition of a composite, such materials generally include such phases as the matrix and the reinforcing phase. The mixture of appropriately selected proportions of these two components proposed by the authors still remains, in my opinion, a composition that can act as a matrix in the composite.
The authors were looking for the most advantageous composition of beeswax and rosin in the context of the application of such material as a natural (ecological) coating of cork plates, which also belong to natural materials. The authors investigated a number of performance properties of the beeswax-rosin composition itself, using different amounts of components (4 different solutions with varying proportions) to determine which blend provides the best performance. The reference material was lime plaster. The adhesion of the composition to cork boards was also tested.
The Introduction section is well structured and properly introduces the subject matter of the authors. The subject matter itself is, in my opinion, extremely important and fits in with the current research trends. Sustainable development issues, including the search for innovative energy-saving and ecological materials, are indicated as the directions of the future, e.g. Building.
The described issues are in line with the subject of the Coatings magazine and have a novelty aspect.
The article is written in understandable language and it is interesting. The number of tests carried out on various properties of the coating and its connection with the cork board is noteworthy. The authors selected the most important properties and examined them. I was not able to find information about the number of samples (the number of repetitions) for a given composition in a given study, but the standard deviations given by the Authors prove that the experiment was repeated for a given composition.
Figures and tables are easy to interpret and understand. In Figure 12, photo is missing (c).
The cited literature is numerous, properly selected and largely up-to-date. In this context, I suggest that you correct the formatting of 38 items of literature, complete the data in items 11 and 31, and check and provide the date of access to the file from 29 items of literature - the link does not work.
The study lacks some Conclusion - such a compact (e.g. in points) summary of the research results.
The authors present among the benefits of the solution the improvement of thermal comfort both in winter and in summer. I missed a bit of research related to this issue. In the future, it may be worth including these issues in the research plan. How will the applied coating affect the conductivity of such a solution? What were the criteria for selecting the thickness of the coating? Visually, the proposed solution seems to be visually unattractive in the context of construction. Interesting effects can be obtained from such a coating in the case of the furniture mentioned by the authors.
Author Response
Response to Reviewer Comments
Dear reviewers,
Thank you very much for giving us the sincere advices to improve our paper. We had carefully read your comments and revised the manuscript accordingly. Noticeably we have the following changes:
Reviewer #2
R: The authors of the article entitled "A Bio Based Render for Insulating Granulated Cork Panels" proposed and tested an innovative external coat for the cork agglomerated panels made of a blending composite of beeswax and rosin (colophony).
At the outset, I have doubts whether the word composite should be used in this case, or rather composition? Although it is difficult to find one consistent definition of a composite, such materials generally include such phases as the matrix and the reinforcing phase. The mixture of appropriately selected proportions of these two components proposed by the authors still remains, in my opinion, a composition that can act as a matrix in the composite.
A: According to reviewer’s suggestions we have replaced the term “composite” with “composition” in all the relevant sections of the article.
R: The authors were looking for the most advantageous composition of beeswax and rosin in the context of the application of such material as a natural (ecological) coating of cork plates, which also belong to natural materials. The authors investigated a number of performance properties of the beeswax-rosin composition itself, using different amounts of components (4 different solutions with varying proportions) to determine which blend provides the best performance. The reference material was lime plaster. The adhesion of the composition to cork boards was also tested.
The Introduction section is well structured and properly introduces the subject matter of the authors. The subject matter itself is, in my opinion, extremely important and fits in with the current research trends. Sustainable development issues, including the search for innovative energy-saving and ecological materials, are indicated as the directions of the future, e.g. Building.
The described issues are in line with the subject of the Coatings magazine and have a novelty aspect.
The article is written in understandable language and it is interesting. The number of tests carried out on various properties of the coating and its connection with the cork board is noteworthy. The authors selected the most important properties and examined them. I was not able to find information about the number of samples (the number of repetitions) for a given composition in a given study, but the standard deviations given by the Authors prove that the experiment was repeated for a given composition.
Figures and tables are easy to interpret and understand. In Figure 12, photo is missing (c).
A: We thank the reviewer for his appreciation of our work. Corrections have been accepted.
R: The cited literature is numerous, properly selected and largely up-to-date. In this context, I suggest that you correct the formatting of 38 items of literature, complete the data in items 11 and 31, and check and provide the date of access to the file from 29 items of literature - the link does not work.
A: References have been revised according to reviewer’s amendments
R: The study lacks some Conclusion - such a compact (e.g. in points) summary of the research results.
A: The conclusion section has been extended and a compact pointed summary of the results obtained has been added, accordingly with the reviewer’s suggestion.
R: The authors present among the benefits of the solution the improvement of thermal comfort both in winter and in summer. I missed a bit of research related to this issue.
A: We explained the concept in page 2 of the manuscript, in particular the cork panels got a higher thermal capacity value than other insulation panel as EPS or XPS this property is important to delay the thermal wave
R: In the future, it may be worth including these issues in the research plan. How will the applied coating affect the conductivity of such a solution?
A: We thank the reviewer for his precious suggestion. We will consider this feature in our future research.
R: What were the criteria for selecting the thickness of the coating?
A: We tried to make clearer the criteria for selecting the thickness of the coating. See page 16,
R: The proposed solution seems to be visually unattractive in the context of construction. Interesting effects can be obtained from such a coating in the case of the furniture mentioned by the authors.
A: This aspect has been taken into account as a future direction of research

Reviewer 3 Report
The paper presents research results of a bio-based thermal insulation material (cork with beeswax and colophony mixture coating). The wok is interesting and a good contribution to the field.
One questions I would like the authors to comment on is that if the proposed coating is free of causing allergies. I am asking this because Colophony allergy well known.
Minor edits:
Line 26: number “1” is missing in “.Introduction”
Please check for inconsistent spacing after a full stop. Example line 29.
Author Response
Response to Reviewer Comments
Dear reviewer,
Thank you very much for giving us the sincere advices to improve our paper. We had carefully read your comments and revised the manuscript accordingly. Noticeably we have the following changes:
Reviewer #3
R:The paper presents research results of a bio-based thermal insulation material (cork with beeswax and colophony mixture coating). The wok is interesting and a good contribution to the field.
A: Thank you for your considerations.
R: One questions I would like the authors to comment on is that if the proposed coating is free of causing allergies. I am asking this because Colophony allergy well known.
A: It is knows that the contact with rosin in an allergic individual provokes acute allergic dermatitis and difficulty breathing. For this reason in this manuscript we proposed the apply of the coating with BR only to the external cork panels to eliminate indoor inhabitants' health risk
R: Minor edits: Line 26: number “1” is missing in “.Introduction”. Please check for inconsistent spacing after a full stop. Example line 29.
A: Thank you, we have revised according to your suggestions
